# Diffusion Barrier Performance of AlCrTaTiZr/AlCrTaTiZr-N High-Entropy Alloy Films for Cu/Si Connect System

**DOI:** 10.3390/e22020234

**Published:** 2020-02-19

**Authors:** Chunxia Jiang, Rongbin Li, Xin Wang, Hailong Shang, Yong Zhang, Peter K. Liaw

**Affiliations:** 1School of Materials Science and Engineering, Shanghai Dianji University, Shanghai 201306, Chinawangxin@sdju.edu.cn (X.W.);; 2State Key Laboratory for Advanced Metals and Materials, University of Science and Technology Beijing, Beijing 100083, China; drzhangy@ustb.edu.cn; 3Department of Materials Science and Engineering, The University of Tennessee, Knoxville, TN 37996, USA; pliaw@utk.edu

**Keywords:** high-entropy alloys, diffusion barriers, thermal stability, amorphous structure, lattice distortion, lattice mismatch

## Abstract

In this study, high-entropy alloy films, namely, AlCrTaTiZr/AlCrTaTiZr-N, were deposited on the n-type (100) silicon substrate. Then, a copper film was deposited on the high-entropy alloy films. The diffusion barrier performance of AlCrTaTiZr/AlCrTaTiZr-N for Cu/Si connect system was investigated after thermal annealing for an hour at 600 °C, 700 °C, 800 °C, and 900 °C. There were no Cu-Si intermetallic compounds generated in the Cu/AlCrTaTiZr/AlCrTaTiZr-N/Si film stacks after annealing even at 900 °C through transmission electron microscopy (TEM) and atomic probe tomography (APT) analysis. The results indicated that AlCrTaTiZr/AlCrTaTiZr-N alloy films can prevent copper diffusion at 900 °C. The reason was investigated in this work. The amorphous structure of the AlCrTaTiZr layer has lower driving force to form intermetallic compounds; the lattice mismatch between the AlCrTaTiZr and AlCrTaTiZ-rN layers increased the diffusion distance of the Cu atoms and the difficulty of the Cu atom diffusion to the Si substrate.

## 1. Introduction

Copper film has a better electron migration resistance and lower resistivity than aluminum film. Copper film is a promising material in the interconnect of very large-scale integration technologies [1]. However, the copper interconnection has the problem of interdiffusion between the copper and silicon, which becomes serious at 180~200 °C [2]. It is necessary to insert a film called a diffusion barrier and adhesion promoter (DBAP) between Cu and Si to increase the failure temperature. With the development of the integrated circuit (IC) industry, the characteristic size has decreased. The conventional barrier materials have microstructure defects.

The diffusion barriers were investigated in recent years, such as the ZrN (20 nm) [3], MoN (50 nm) [4], TaN (8 nm) [5], and Ir/TaN (5/5 nm) [6]. Most of these diffusion barriers contain two or three elements. These traditional diffusion barrier materials cannot block the diffusion of the Cu and Si at about 600 °C, which greatly limits the development of ICs. The materials with more elements in various material systems may be a good choice [7,8,9]. One of the attempts is based on multi-element materials, involves not just more elements, but more major elements, which was called high-entropy alloys (HEAs) by Yeh and co-workers [10,11,12,13,14,15,16,17]. HEAs have four main core effects: (1) thermodynamic, high-entropy effects; (2) kinetically-sluggish diffusion; (3) structurally, severe lattice distortion; and (4) cocktail effect properties [18,19]. According to prior studies [20,21,22,23,24], high-entropy alloys have great potential for coating applications. The amorphous structures develop easily in high-entropy alloys [16]. They have the thermal stability of high-entropy alloys [25] and chemical stability after high temperature annealing [26]. These characteristics of high-entropy alloys and their nitride films are important advantages for using high-entropy alloys as diffusion barriers. High-entropy alloys can be candidates for next-generation diffusion barriers [27,28]. A previous study showed that HEAs, such as the diffusion barrier of AlMoNbSiTaTiVZr50N50 [27], can sustain a high temperature of 800 °C for 30 min. The sandwiched Cu/FeCoBTiNb/Si [29] film showed a failure temperature of 550 °C for 30 min by inhibiting the Cu diffusion. The (TiTaCrZrAlRu)N [30] film can sustain a high temperature of 900 °C for 30 min. Compared with traditional diffusion barriers, the high-entropy alloy diffusion barriers have a higher failure temperature. However, the problem for the bonding of the diffusion barriers and the Cu film is not mentioned. In this paper, not only is the diffusion barrier performance of the diffusion barriers is studied, but also the bonding between the Cu film and the diffusion barrier layer are studied.

In this study, the AlCrTaTiZr/AlCrTaTiZr-N (HEA/HEAN) films were inserted between the Cu layer and Si substrate for promoting bonding, electrical characteristics, and thermal stability of the Cu/AlCrTaTiZr/AlCrTaTiZr-N/Si film stacks. Its microstructures and properties were investigated by the cross-sectional transmission electron microscopy (XTEM) and X-ray diffraction (XRD). Its electrical characteristics were investigated by a four-point probe station (4PP). The results show that AlCrTaTiZr/AlCrTaTiZr-N films can prevent the Cu and Si diffusion.

## 2. Experimental Details

The AlCrTaTiZr/AlCrTaTiZr-N films were deposited on Si substrates by the UDP 650/4 magnetron sputtering equipment. The size of the sputtering target used was length (380 mm) by width (175 mm) by thickness (10 mm). The sputtering target contained equimolar Al, Cr, Ta, Ti, and Zr elements, which were obtained by vacuum arc melting.

Acetone and ethanol were used to clean the Si substrate before depositing the HEAs films. The parameters setting for AlCrTaTiZr deposition on Si substrates was 0.8 A current, −80 V bias voltage, room temperature, and 20 sccm Ar gas. The parameter setting for AlCrTaTiZr-N deposition on AlCrTaTiZr layer was 0.8 A current, −80 V bias voltage, room temperature, and mixed gases with 16 sccm Ar and 4 sccm N2. The chamber vacuum pressure was 2.5 × 10^−5^ Torr before sputtering. The working pressure was 2 × 10^−3^ Torr during sputtering. The AlCrTaTiZr/AlCrTaTiZr-N films thickness was about 20 nm.

The Cu film was deposited on the AlCrTaTiZr-N layer by the UDP 650/4 magnetron sputtering equipment at a current of 1 A, a bias voltage of −50 V, room temperature, and 20 sccm Ar gas. The chamber vacuum pressure was 2.5 × 10^−5^ Torr before sputtering. The working pressure was 2 × 10^−3^ Torr during sputtering. The film deposition time was six minutes, and the film thickness was about 200 nm.

Then, the Cu/AlCrTaTiZr/AlCrTaTiZr-N/Si film stacks were annealed in an encapsulated vacuum quartz tube with pressure <1 × 10^−5^ Torr for an hour at 600 °C, 700 °C, 800 °C, and 900 °C. In order to avoid the delamination from thermal stresses, a cooling rate of 5 °C/min was used once the temperature was above 200 °C. Below this temperature, natural cooling in the air for the film stacks was allowed.

The surface morphology of the as-deposited and thermally annealed Cu/AlCrTaTiZr/AlCrTaTiZr-N/Si film stacks was observed by the scanning electron microscope (SEM) in a Nova Nano (FEI, Hillsboro, Oregon, USA) with an acceleration voltage of 15 kV. The X-ray diffractometer (XRD) with a D8 Advanced (Bruker, Billerica, MA, USA) was installed with Cu Kα radiation (tube parameters: 40 kV; 40 mA). The diffractograms were measured in a diffraction angle range (2θ) of 20° to 80° with a step size of 0.02° and 0.1 s/step. The crystal structures of the film stacks were analyzed by the XRD. The analysis of the overall chemical composition was carried out by energy dispersive X-ray spectroscopy (EDS) with an Apollo XP spectrometer (EDAX, Philadelphia, PA, USA). The sheet resistance of the as-deposited and thermally annealed film stacks was measured by four-point probe with a Model 280SI (4D, America). The microstructures and lattice structures of the as-deposited and annealed films were observed by a high-resolution transmission electron microscope (HR-TEM) with a H-600 (Hitachi, Tokyo, Japan). The 3D atomic distribution within Cu/AlCrTaTiZr/AlCrTaTiZr-N/Si was measured by the atomic probe tomography (APT) with a LEAP4000XHR (CAMECA, Paris, France).

## 3. Results

The results of the EDS analysis (Table 1) revealed that the contents of metallic elements in the AlCrTaTiZr layer and AlCrTaTiZr-N layer were nearly equimolar ratio and close to the ratio of metallic elements in the sputtering target.

Figure 1 shows the XRD patterns of the AlCrTaTiZr layer and AlCrTaTiZr-N layer. Figure 1 revealed a broad peak in the AlCrTaTiZr layer, indicating that this layer presented an amorphous structure. High-entropy alloys have high mixed entropy, and the atomic size difference of metal atoms will cause severe lattice distortion, which promotes the formation of amorphous structures in thin films [31]. It can be seen that four diffraction peaks at 36.5°, 42°, 59.5°, and 78.4° corresponded to the (111), (200), (220), and (222) lattice planes of the face centered cubic (FCC) AlCrTaTiZr-N layer. The AlCrTaTiZr-N layer had FCC crystal structures rather than co-existing separated nitrides [32].

The SEM images of the Cu/AlCrTaTiZr/AlCrTaTiZr-N/Si film stacks are shown in Figure 2. It shows the cross-sectional morphologies of the as-deposited, 600 °C, 800 °C, and 900 °C annealed Cu/AlCrTaTiZr/AlCrTaTiZr-N/Si film stacks. For the as-deposited film stacks, the interface between the Cu and AlCrTaTiZr layers was clear. There was no gap. After annealing at 600 °C and 800 °C, the contact between the AlCrTaTiZr and Cu layers was still good. The Cu layer did not peel off, and there was no gap between the AlCrTaTiZr and Cu layers. After annealing at 900 °C, gap and holes could be observed in the Cu/AlCrTaTiZr/AlCrTaTiZr-N/Si film stacks. However, the Cu layer still did not peel off.

Figure 3 shows the SEM surface morphologies of the as-deposited, 600 °C, 800 °C, and 900 °C annealed Cu/AlCrTaTiZr/AlCrTaTiZr-N/Si film stacks. Figure 3a shows that the as-deposited copper layer was smooth and had no obvious characteristics on its surface. Figure 3b shows that the copper layer had copper grain growth on the surface after annealing at 600 °C. Grain growth happened on the copper layer with a diameter 40 nm after 600 °C, 0.5 μm after 700 °C, and 1 μm after 800 °C annealing. Figure 3d shows there were a little of micro-holes in the copper layer after 900 °C annealed Cu/AlCrTaTiZr/AlCrTaTiZr-N/Si film stacks. The formation of these micro-holes might be due to the thermal stress of the copper layer, rather than the diffusion of copper.

Figure 4 shows the XRD patterns of the as-deposited, 600 °C, 800 °C, and 900 °C annealed Cu/AlCrTaTiZr/AlCrTaTiZr-N/Si film stacks. For as-deposited film stacks, three diffraction peaks at 43.4°, 50.6°, and 74.3° corresponded to the (111), (200), and (220) lattice planes of the FCC Cu. The diffraction peaks of the copper layer did not change after 600 °C, 800 °C, and 900 °C annealing. Grain growth in the copper layer was responsible for the sharper diffraction peaks. Figure 4 reveals a broad peak in the analysis, indicating that this film presented an amorphous structure. The AlCrTaTiZr metallic layer was an amorphous structure [33]. After 900 °C annealing, some compounds or solution phases were formed due to the reactions between the AlCrTaTiZr-N layer and Si substrate or between the AlCrTaTiZr-N layer and Cu layer, such as Cr4Si4Al13. Although the Cu/AlCrTaTiZr/AlCrTaTiZr-N/Si film stacks were annealed at even 900 °C, the Figure 4 still exhibits no signals of copper silicide which indicated that the AlCrTaTiZr/AlCrTaTiZr-N films still acted as an effective barrier for the interdiffusion of Si and Cu at a high temperature.

Figure 5 shows the sheet resistance of the as-deposited, 600 °C, 800 °C, and 900 °C annealed Cu/AlCrTaTiZr/AlCrTaTiZr-N/Si film stacks to identify the interdiffusion of Si and Cu. As plotted in Figure 5 clearly, the sheet resistance decreased from 0.248 Ω/square to 0.058 Ω/square after 600 °C annealing. The reason is that grain growth of the Cu layer will eliminate most grain boundaries and defects. At 900 °C, the sheet resistance of the film stacks increased to approximately 4.62 Ω/square. However, according to the aforementioned XRD and SEM analyses, the resistivity increase was possibly not due to the failure of the film stacks since the resistivity would drastically increase once the Cu silicides formed.

## 4. Discussion

As shown in the cross-sectional morphologies in Figure 2, the gap appeared due to the different coefficients of thermal expansion for the metal nitrides and Cu. The coefficient of thermal expansion for Cu is 17.5 × 10^−6^ m/°C which is larger than the metal nitride. For example, the coefficient of thermal expansion for TiN is 6.8 × 10^−6^ m/°C and for TaN is 4.2 × 10^−6^ m/°C [14]. If the Cu layer is directly sputtered on the metal nitride layer surface, the thermal expansion amount between them are significantly different during high temperature annealing. This difference caused a poor bonding property between the AlCrTaTiZr-N layer and Cu layer, resulting in a peeling off tendency of the Cu layer [34,35].

The coefficients of thermal expansion for the AlCrTaTiZr and AlCrTaTiZr-N layers were calculated. The coefficients of thermal expansion for each element in AlCrTaTiZr are listed as follows: Al: 27.4 × 10^−6^ m/°C, Cr: 6.2 × 10^−6^ m/°C, Ta: 7.0 × 10^−6^ m/°C, Ti: 10.8 × 10^−6^ m/°C, and Zr: 6.9 × 10^−6^ m/°C [36]. According to the following formula [35]
(1)$$\alpha =\sum \alpha_{i}\times i\%$$
where *α* is the coefficient of thermal expansion for AlCrTaTiZr alloy, *α_i_* is the coefficient of thermal expansion for each element, and *i%* is the ratio for each element in the AlCrTaTiZr layer. The sputtering target was prepared with an equimolar ratio of elements. The coefficient of thermal expansion for the AlCrTaTiZr layer is approximately 11.66 × 10^−6^ m/°C and close to the thermal expansion coefficient of Cu. Thus, sputtering the AlCrTaTiZr layer on the AlCrTaTiZr-N layer can enhance the bonding between Cu and AlCrTaTiZr-N layer.

The 700 °C and 800 °C annealed Cu/AlCrTaTiZr/AlCrTaTiZr-N/Si film stacks had lower sheet resistance than that of the as-deposited film stacks. Heat treatment effectively eliminated the defects in the copper layer. However, grain growth and some impurities, such as oxygen and nitrogen, could reduce the electron scattering, thereby reducing the resistance of the film stacks.

The results of XRD, SEM, and 4PP only roughly revealed the crystal and structural changes in the film stacks. Our interest was focused on the effect of the annealing temperature on the microstructure changes of the film stacks. The HR-TEM cross-sectional morphology of the Cu/AlCrTaTiZr/AlCrTaTiZr-N/Si film stacks are shown in Figure 6. Before and after annealing, an approximately 3 nm SiO_2_ layer was clearly observed between the Si and AlCrTaTiZr-N layers. In Figure 6a, four layers including the SiO_2_ layer, AlCrTaTiZr-N layer, AlCrTaTiZr layer, and Cu layer could be clearly identified before annealing. The AlCrTaTiZr layer was characterized as an amorphous structure. A previous article indicated that the formation of amorphous structures is mainly caused by the different atomic size of the elements in high-entropy alloys [31]. Moreover, the lattice spacings of these nanocrystalline phases were determined to be approximately 0.241 nm and 0.218 nm, respectively, which are similar to the average values of the (111) and (220) planes of the mixed FCC structure in Table 2. After 900 °C annealing, crystallization of the amorphous structure was observed. The measured lattice spacing of the nanocrystalline phase is approximately 0.213 nm, which is similar to the average values of the (111) planes of Al, Cr, Ta, Ti, and Zr (Table 3). As exhibited in Figure 6c, unclear interfaces among the Cu layer, AlCrTaTiZr layer, and AlCrTaTiZr-N layer formed after 900 °C annealing. However, the interfaces among the AlCrTaTiZr-N layer, SiO_2_ layer, and Si layer were still clearly distinguished. As observed, some intermetallic compounds appeared, such as a Cu–AlCrTaTiZr phase, around the AlCrTaTiZr/Cu interface. Intermetallic compounds were also observed in the AlCrTaTiZr/AlCrTaTiZr-N films. Si–AlCrTaTiZr-N phase was observed around the AlCrTaTiZr-N/Si interface [37]. During 900 °C annealing, Si atoms were diffused into the AlCrTaTiZr-N layer. However, the diffusion of Cu and Si was retarded by the AlCrTaTiZr/AlCrTaTiZr-N barrier films. The lattice structure of the Si substrate was still observed, and no signals of any other Cu–Si phases are found, which is consistent with the XRD and 4PP results.

Figure 7 shows the distribution of the Cu/AlCrTaTiZr/AlCrTaTiZr-N alloy films elements (Cu, Al, Cr, Ta, Ti, Zr, and N) after annealing at 900 °C, which was measured by APT. As observed, the composition of the AlCrTaTiZr/AlCrTaTiZr-N films were relatively uniform after annealing at 900 °C. The layer had no obvious crystal boundary and partial clustering. The Cu layer had some diffusion into the AlCrTaTiZr layer with a distance of less than 10 nm, which is consistent with the HR-TEM results shown in Figure 6e. The Cu diffusion into the Si substrate was apparently retarded by the AlCrTaTiZr/AlCrTaTiZr-N barrier films.

According to the above SEM, HR-TEM, XRD, 4PP, and APT results, the AlCrTaTiZr/AlCrTaTiZr-N films have an excellent diffusion barrier performance. The superior performance of the AlCrTaTiZr/AlCrTaTiZr-N films should be attributed to its excellent structural and chemical stability. The insert of an amorphous layer is the major factor. The grain boundaries of the diffusion barrier are fast diffusion pathways of Cu atoms. The stability of the amorphous barrier structure can avoid this problem.

At first, sputtering the amorphous layer can inhibit the activity of oxygen, which can prevent the oxidation of the diffusion barrier layer. If the diffusion barrier layer was oxidized after high temperature annealing, oxygen can diffuse through the grain boundary of the Cu to the surface of the Cu layer. This leads to the agglomeration of Cu and causes the Cu layer to break [35]. Additionally, if the oxygen content is high in the film stacks, the electron scattering of the film is relatively large which will increase the square resistance of the film stacks. After annealing, the oxides will weaken the electrical and physical properties of the Cu/AlCrTaTiZr/AlCrTaTiZr-N/Si film stacks. Therefore, the AlCrTaTiZr layer inhibits the activity of oxygen, which can improve the diffusion barrier performance of the films and increase the reliability of the Cu interconnection [37].

Secondly, the amorphous structure of the AlCrTaTiZr layer has a lower driving force to form intermetallic compounds. The driving force can be estimated by the formation energy. The formation energy (ΔG) of the AlCrTaTiZr alloy can be expressed as:(2)ΔG=ΔH−TΔS
where the ΔH is the formation enthalpy and ΔS is the formation entropy. The ΔH formation enthalpy of AlCrTaTiZr can be calculated by the following Equation (3) [38]:(3)$$\Delta H=4\sum_{\begin{array}{l} i,j=1\\ i>j \end{array}}^{n}\Delta H_{ij}^{mix}c_{i}c_{j}$$
where *n* is the total number of elements in the alloy, *c_i_* and *c_j_* are the concentration of the *i* and *j* elements, and the $\Delta H_{ij}^{mix}$ is the mixing heat of the *i* and *j* elements. Table 3 is the values of $\Delta H_{ij}^{mix}$ [39]. Accordingly, the calculated the formation enthalpy (ΔH) is −20 kJ/mol.

The formation entropy can be calculated according to the Boltzmann’s entropy formula:(4)$$\Delta S_{}=k\ln\omega$$
(5)$$k=\frac{R}{Na}$$
where *k* is the Boltzmann’s constant, *ω* is the number of ways of mixing, *R* is the gas constant, and *Na* is the Avogadro’s constant. Therefore, the configurational entropy of the mixed AlCrTaTiZr alloy is 13.38 J/(mol·K). The formation energy of AlCrTaTiZr is −42.11 kJ/mol at 900 °C. It can be seen in Table 4 that these elements alloy enthalpy of mixing with respect to each other have approaching zero or positive. The result means that the Al, Cr, Ta, Ti, and Zr react with each other difficultly [14].

The severe lattice distortion is the third factor that contributes to the stability of the amorphous structure. The severe lattice distortion impedes the crystallization of the AlCrTaTiZr layer. The severe lattice distortion in the AlCrTaTiZr layer results from the large atomic size differences among the Al, Cr, Ta, Ti, and Zr atoms. Lattice distortion causes strain energy. The amorphous structure has no lattice framework, so the strain energy of the amorphous structure can be regarded as zero. It has been pointed out that amorphous metals whose constituent elements show large size differences have a higher atomic packing density [40]. This trend means that it is difficult for these metal atoms in the amorphous structure to be rearranged, and these atoms have less free volume.

Fourthly, the lattice mismatch between the AlCrTaTiZr and AlCrTaTiZr-N layers increased the diffusion distance of the Cu atoms, thereby increasing the difficulty of Cu atoms diffusion into the Si substrate. In Figure 8, the diffusion pathways of Cu atoms in the AlCrTaTiZr-N diffusion barrier structures and the AlCrTaTiZr/AlCrTaTiZr-N diffusion barrier structures are compared. The atoms in the AlCrTaTiZr layer at the grain boundary of AlCrTaTiZr-N layer can block the diffusion pathways of the Cu atoms.

As shown in Figure 8b, the insert of the AlCrTaTiZr layer increases the diffusion distance of the Cu atom on the barrier layer, according to the diffusion equation [30].
(6)$$X=2\sqrt{Dt}$$
where *X* is the diffusion length, *D* is the diffusion coefficient, and *t* is the diffusion time. According to the above study, the diffusion time is increased as the diffusion distance is increased with the alloy films inserted. Then, diffusion coefficient *D* is decreased as *X* is not changed [30]. The diffusion coefficient equation is
(7)$$D=D_{0}\exp(-\frac{Q_{a}}{RT})$$
where *D*_0_ is the diffusion constant, *Qa* is the diffusion activation energy of atomic, *R* is the gas constant, and *T* is the thermodynamic temperature. Without consideration of the diffusion constant *D*_0_, the diffusion activation energies of atoms are valid both for the “thin film” and “semi-infinite” models [41]. As shown in the formula, the diffusion coefficient is inversely proportional to the diffusion activation energy while the other parameters remain constant. Therefore, a decrease in the diffusion coefficient of Cu atoms in the diffusion barrier films will increase the difficulty of Cu atoms diffusion in the diffusion barrier films [42,43].

## 5. Conclusions

The AlCrTaTiZr/AlCrTaTiZr-N films were investigated in this study as diffusion barrier for Cu interconnects. The Cu/AlCrTaTiZr/AlCrTaTiZr-N/Si film stacks were annealed at 900 °C for an hour. The interface between the AlCrTaTiZr/AlCrTaTiZr-N films and Si substrate was clear. The bonding property of the Cu and AlCrTaTiZr-N layer could be improved by inserting an AlCrTaTiZr layer. Even at 900 °C, the AlCrTaTiZr/AlCrTaTiZr-N films with a thickness of 20 nm had a good resistance to the Cu and Si interdiffusion. The amorphous structure of the AlCrTaTiZr layer had a lower driving force to form intermetallic compounds. The severe lattice distortion and the reduced diffusion kinetics were the factor for the AlCrTaTiZr/AlCrTaTiZr-N films to act as a very effective barrier material for inhibiting the diffusion of the Cu. The lattice mismatch of the AlCrTaTiZr and AlCrTaTiZr-N layers increased the diffusion distance and the difficulty of the Cu atom diffusion to the Si substrate. All the above results indicated that the AlCrTaTiZr/AlCrTaTiZr-N films have an excellent diffusion barrier performance.

## Figures and Tables

**Figure 1 entropy-22-00234-f001:**
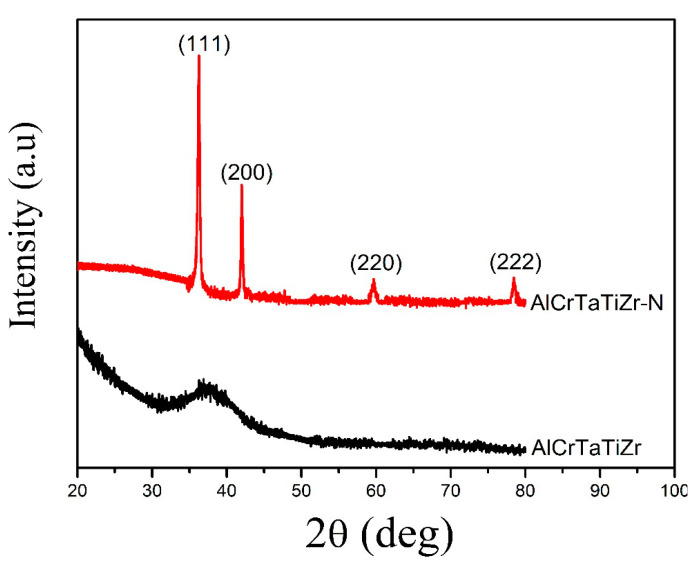
XRD patterns of the AlCrTaTiZr layer and AlCrTaTiZr-N layer.

**Figure 2 entropy-22-00234-f002:**
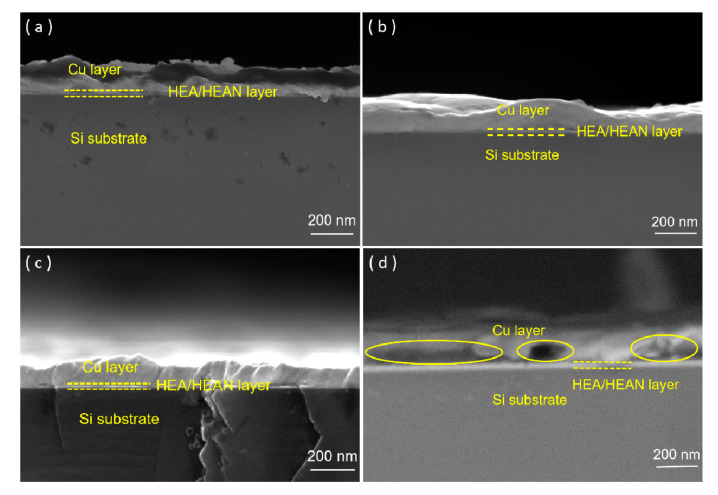
SEM cross-sectional morphologies of (**a**) as-deposited, (**b**) 600 °C annealed, (**c**) 800 °C annealed, and (**d**) 900 °C annealed Cu/AlCrTaTiZr/AlCrTaTiZr-N/Si film stacks.

**Figure 3 entropy-22-00234-f003:**
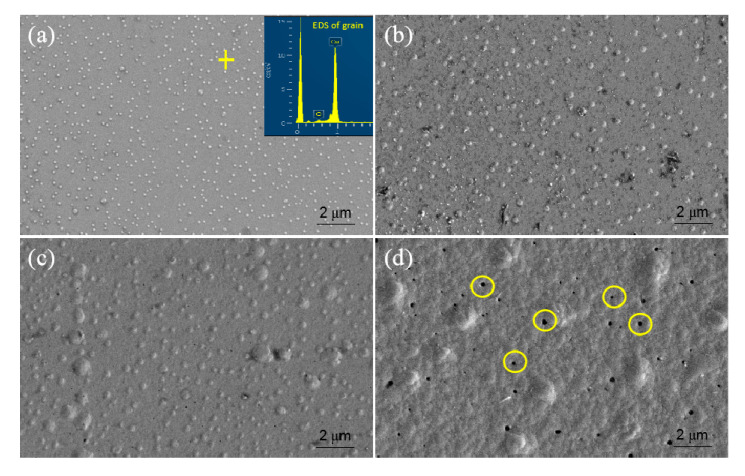
SEM surface morphologies of (**a**) as-deposited, (**b**) 600 °C annealed, (**c**) 800 °C annealed, and (**d**) 900 °C annealed Cu/AlCrTaTiZr/AlCrTaTiZr-N/Si film stacks.

**Figure 4 entropy-22-00234-f004:**
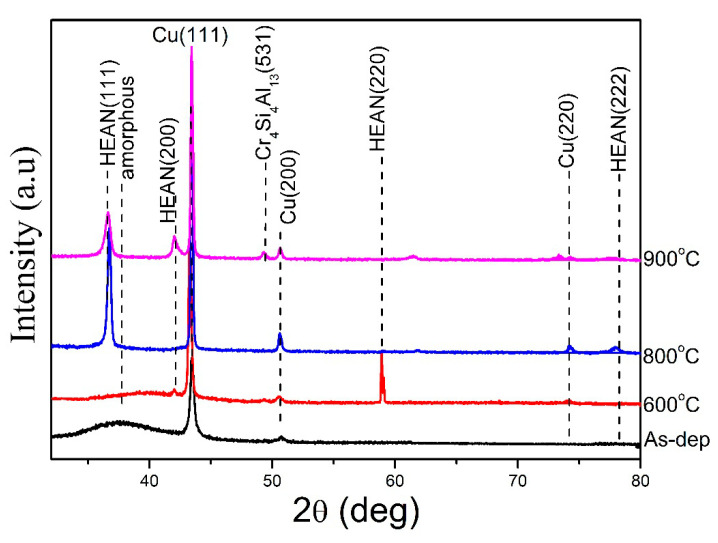
XRD patterns of the as-deposited, 600 °C annealed, 800 °C annealed, and 900 °C annealed Cu/AlCrTaTiZr/AlCrTaTiZr-N/Si film stacks.

**Figure 5 entropy-22-00234-f005:**
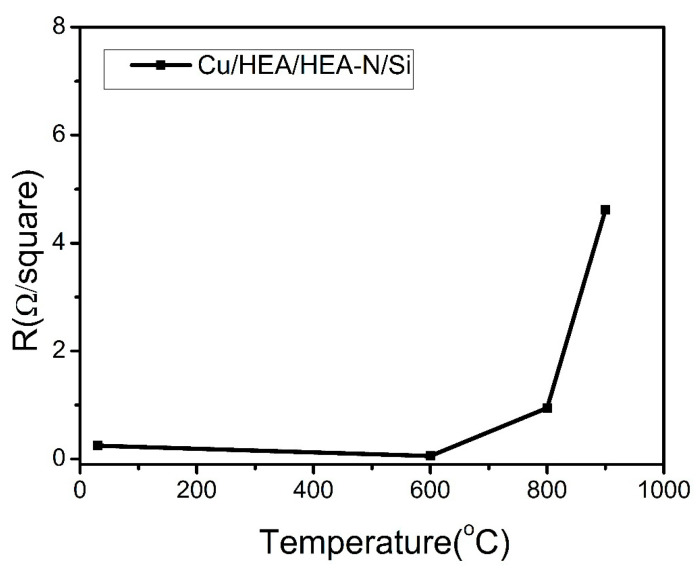
The sheet resistance of the Cu/AlCrTaTiZr/AlCrTaTiZr-N/Si film stacks at different annealing temperatures.

**Figure 6 entropy-22-00234-f006:**
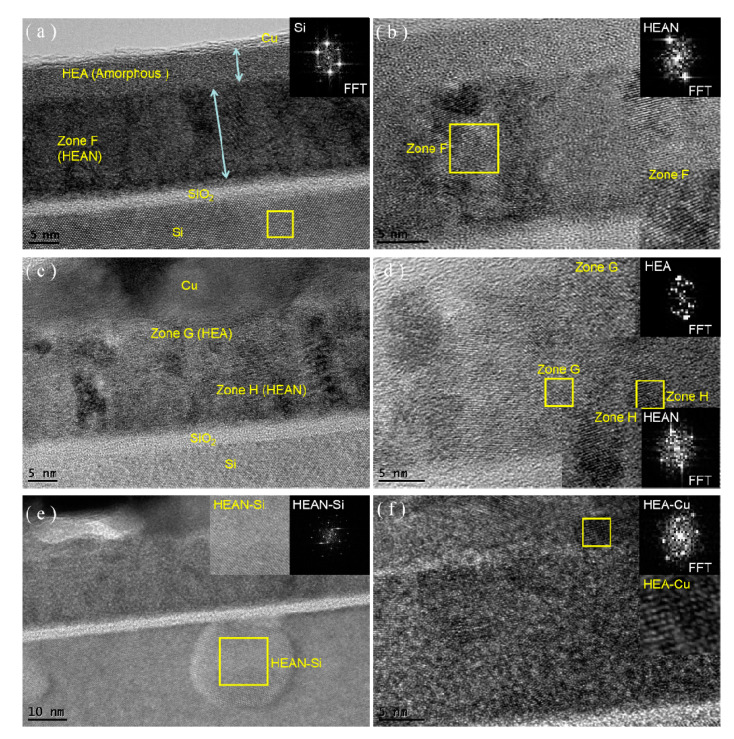
(**a**) cross-sectional high-resolution transmission electron microscope (HR-TEM) image of the as-deposited Cu/AlCrTaTiZr/AlCrTaTiZr-N/Si film stacks; (**b**) corresponding HR-TEM image of the zone C, and (**c**)–(**f**) cross-sectional HR-TEM image of the 900 °C annealed Cu/AlCrTaTiZr/AlCrTaTiZr-N/Si film stacks.

**Figure 7 entropy-22-00234-f007:**
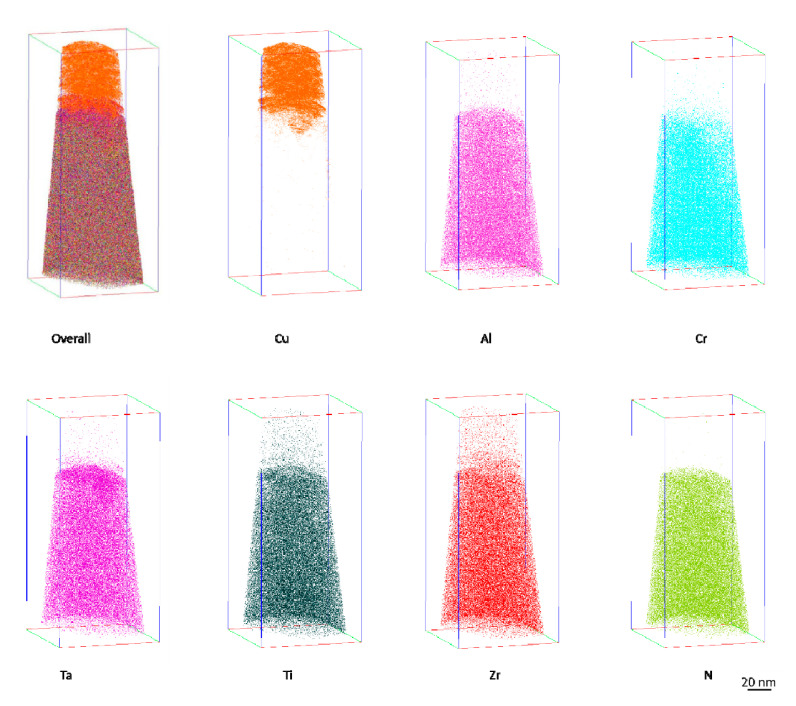
Atomic probe tomography (APT) elemental maps presenting the Cu/AlCrTaTiZr/AlCrTaTiZr-N/Si film stacks with atomic positions of the Cu, Al, Cr, Ta, Ti, Zr, and N.

**Figure 8 entropy-22-00234-f008:**
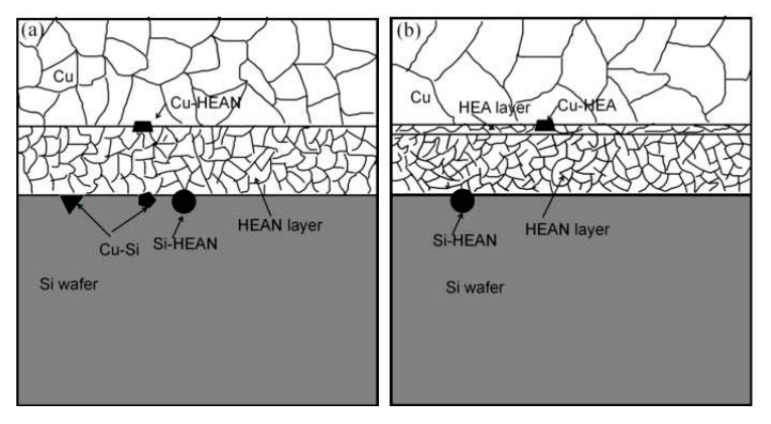
(**a**) the diffusion pathways of the AlCrTaTiZr-N structure and (**b**) the diffusion pathways of the AlCrTaTiZr/AlCrTaTiZr-N structure.

**Table 1 entropy-22-00234-t001:** Composition of the AlCrTaTiZr layer and AlCrTaTiZr-N layer (At%).

	Al	Cr	Ta	Ti	Zr	N
Sputtering target	21.28	20.83	19.86	19.11	18.92	0
AlCrTaTiZr	22.15	22.56	18.28	19.46	17.54	0
AlCrTaTiZr-N	16.64	17.22	15.31	15.11	14.69	21.02

**Table 2 entropy-22-00234-t002:** Crystal structures and lattice constants of the individual metallic elements in AlCrTaTiZr-N.

Nitride	AlN	CrN	TaN	TiN	ZrN	HEAN (Average)
Crystal structure	FCC	FCC	FCC	FCC	FCC	
d(111) (nm)	0.247	0.241	0.247	0.252	0.260	0.248
d(220) (nm)	0.216	0.208	0.218	0.223	0.229	0.218

**Table 3 entropy-22-00234-t003:** Crystal structures and lattice constants of the individual metallic elements in AlCrTaTiZr.

Metal	Al	Cr	Ta	Ti	Zr	HEA (Average)
Crystal structure	FCC	BCC	BCC	HCP	HCP	
d(111) (nm)	0.216	0.208	0.218	0.223	0.229	0.218

**Table 4 entropy-22-00234-t004:** Values of Hmix (kJ/mol) for all possible atomic pairs in the AlCrTaTiZr-N alloy.

	N	Al	Cr	Ta	Ti	Zr	Cu	Si
N	___	−92	−107	−173	−190	−233	−84	−81
Al	___	___	−10	−19	−30	−44	−1	−19
Cr	___	___	___	−7	−7	−12	12	−37
Ta	___	___	___	___	1	3	−17	−55
Ti	___	___	___	___	___	0	15	−66
Zr	___	___	___	___	___	___	−23	−84
Cu	___	___	___	___	___	___	___	−19

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
