# Peer review of "Diffusion Barrier Performance of AlCrTaTiZr/AlCrTaTiZr-N High-Entropy Alloy Films for Cu/Si Connect System"

_entropy, 2020, doi:10.3390/e22020234_

Round 1

Reviewer 1 Report

A high entropy alloy thin film was suggested and demonstrated to work as a diffusion barrier film. The novelty of the topic may merit the publication of this manuscript, however, some issues should be resolved before the publication.

(major) English writing should be significantly improved. Probably the authors may want to consult to some professionals to improve the writing. Without the significant improvement, the manuscript cannot be published.

(major) The high entropy alloy film deposited under nitrogen pressure appears the key foundation of this study. However, the identification of this HEA-N film does not seem to be provided. Is it in the form of nitride like other diffusion barrier films ? Or, nitrogen simply in the interstitial sites of HEA ? The crystallographic information of HEA and HEA-N films should be clearly provided.

Author Response

Dear Professor,

Thanks for you useful comments and suggestions. I read them one by one carefully. And items below are all impvoed. Please see the attachments for details.

English language and style are extensively modified.
Introduction is modified.
Reserach design is modified. Methods are described with informations. Results are cleared as you suggest. Conclusions are modified to support the results.

Looking forward to your feedback.

Best regards

Rongbin Li

Reviewer 2 Report

This paper reports the diffusion barrier performance of AlCrTaTiZr/AlCrTaTiZr-N (HEA/HEAN) films for Cu/Si connect system. The authors investigated the morphologies of system with as-deposited and annealed under several conditions. They have found that amorphous HEA causes the difficulty of Cu diffusion into the Si substrate.

In the large-scale integration technology, the interdiffusion of Cu is serious problem. This paper suggests that HEA film might be one of the solutions. Nowadays HEA research is growing interest in the materials science. The paper is well organized and stimulates researchers in the field of HEA. So I believe the manuscript meets all criteria necessary for Entropy. But, before the acceptance, I recommend the authors to address the comments listed below in order to improve the readability.

The authors mentioned the diffusion barrier materials such as TiN, MoN, W2N, TaN, WNx, TiW and Ir/TaN in the Introduction. The comparison of diffusion barrier performance between these non HEA films and the present one should be discussed. Also, the comparison of diffusion barrier performance between the previous HEA film (ref [30]) and the present one should be discussed. In Fig. 1 (a), the HEA/HEAN film seems to be flat, however the Cu film shows the rather rough surface. Why does the Cu film show such a rough surface? In Fig. 2(a), the bright grains are observed. What are these? The authors should give the film thickness of HEA/HEAN. Figure 4 should be expanded, or the log scale should be employed. In Fig. 5, the direction of film stacking of (a) is different from that of (c). I recommend the authors to use the same directions in all the figures. There are misspellings and grammatical errors. For example, in line 14 “than” would be “then”, in line 21 “investigate” would be “investigated” and in line 251 “resules” would be “results”. I recommend the authors to use an English editing company in the revision of manuscript.

Author Response

Dear Professors,

Thanks for you useful comments and suggestions. I read them one by one carefully. And items below are all impvoed. Please see the attachments for details.

English language and style are extensively modified. Results are cleared as you suggest.

Looking forward to your feedback.

Best regards

Rongbin Li

Round 2

Reviewer 1 Report

The authors successfully revised the manuscript and, therefore, the manuscript can be accepted for publication.